# Quantum latent distributions in deep generative models

Omar Bacarreza [1]  Thorin Farnsworth [1]  Alexander Makarovskiy [1]  Hugo Wallner [1]  Tessa Hicks [1]
Santiago Sempere-Llagostera [1]  John Price [1]  Robert J. A. Francis-Jones [1]  William R. Clements [1]

## Abstract

Many successful families of generative models leverage a low-dimensional latent distribution that is mapped to a data distribution. Though simple latent distributions are often used, the choice of distribution has a strong impact on model performance. Recent experiments have suggested that the probability distributions produced by quantum processors, which are typically highly correlated and classically intractable, can lead to improved performance on some datasets. However, when and why latent distributions produced by quantum processors can improve performance, and whether these improvements are connected to quantum properties of these distributions, are open questions that we investigate in this work. We show in theory that, under certain conditions, these "quantum latent distributions" enable generative models to produce data distributions that classical latent distributions cannot efficiently produce. We provide intuition as to the underlying mechanisms that could explain a performance advantage on real datasets. Based on this, we perform extensive benchmarking on a synthetic quantum dataset and the QM9 molecular dataset, using both simulated and real photonic quantum processors. We find that the statistics arising from quantum interference lead to improved generative performance compared to classical baselines, suggesting that quantum processors can play a role in expanding the capabilities of deep generative models.

## 1. Introduction

In recent years, several successful families of deep generative models have emerged that learn to map samples from a latent distribution, a typically low-dimensional probability distribution, to a higher-dimensional data distribution. For instance, models that use latent distributions such as generative adversarial networks (GANs) (Goodfellow et al., 2014; Karras et al., 2019), latent diffusion models (Rombach et al., 2022; Blattmann et al., 2023), and flow matching models (Lipman et al., 2023; Albergo & Vanden-Eijnden, 2023) have all achieved remarkable performance on a wide range of datasets from images to proteins. During training, these models learn a mapping between the structure of the latent distribution and that of the data. As such, the structure of this latent distribution plays a critical role. For instance, in both GANs (Arici & Celikyilmaz, 2016; Karras et al., 2019; Hu et al., 2023) and flow matching models (Li et al., 2024; Lee et al., 2026), matching the structure of the latent distribution used by the generator to that of the data has been shown to significantly improve performance.

The design space for these algorithms has typically been limited to the range of functions that are efficient to implement on modern CPUs and GPUs. As such, relatively simple latent distributions are typically used, for example a multivariate Gaussian distribution transformed by a neural network (Karras et al., 2019). However, the use of simple latent distributions with finite-capacity neural network architectures can be a limiting factor for modeling complex datasets (Hu et al., 2023). As an example, many quantum processes cannot be efficiently simulated using classical (i.e. non-quantum) computational methods (Aaronson & Arkhipov, 2011). Learning the data distribution arising from such a process is thus likely to be challenging for classical generative models using a simple latent distribution.

Quantum computers have been proposed as a way to broaden the range of latent distributions that can be efficiently produced (Rudolph et al., 2022). This is motivated by recent progress in quantum computing technology, with several quantum computers now competitive against classical supercomputers (Arute et al., 2019; Madsen et al., 2022) for some distribution sampling tasks (Hangleiter & Eisert, 2023). Using a quantum computer to produce samples from a non-classical latent distribution has yielded promising empirical results on a wide range of models and datasets (Rudolph et al., 2022; Wilson et al., 2021; Li et al., 2021; Kao et al., 2023; Xiao et al., 2024; Jin & Merz Jr, 2025). However, the lack of a theoretical understanding of how

[1]ORCA Computing, London, UK. Correspondence to: William Clements <wclements@orcacomputing.com>.

*Proceedings of the $43^{rd}$ International Conference on Machine Learning*, Seoul, South Korea. PMLR 306, 2026. Copyright 2026 by the author(s).

a quantum distribution can affect model performance and the limited benchmarking performed to date have made it difficult to evaluate this approach.

In this work, we develop a deeper understanding of the use of quantum latent distributions, and perform experiments to benchmark this approach. We theoretically investigate the relationship between the complexity class of the latent distribution and that of the generated distribution. We show that, under some assumptions on the neural network architecture, a quantum latent distribution can produce a wider range of distributions in the data space than could be achieved with classical latent distributions. This can allow a model to achieve improved performance on some datasets. We develop an intuition for why and when this improved performance can be achieved in practice. Based on this intuition, we select a synthetic quantum dataset and the QM9 chemistry dataset for benchmarking experiments. We compare a quantum distribution produced by quantum interference between photons to a range of classical distributions, including a photonic distribution where we switched off quantum interference effects. We find that the quantum distribution outperforms the other distributions on these datasets, indicating that some properties of quantum distributions can indeed lead to improved generative performance.

Our main contributions are as follows:

- We show that, under some assumptions on the generative model, using a quantum distribution in the latent space allows the generative model to produce distributions that could not be efficiently produced classically.

- We provide intuition as to when and why a quantum distribution can lead to improved performance on a real dataset.

- We benchmark this approach on both a synthetic quantum dataset and the QM9 quantum chemistry dataset, and show that a quantum latent distribution in a GAN can improve model performance compared to a range of relevant classical distributions.

While performance advantages yielded by a quantum latent distribution are dataset and model dependent, our results validate the notion that these distributions can be a useful tool to improve the performance of deep generative models.

*Conflict of interest disclosure: The authors are employed by ORCA Computing, which commercializes the type of quantum processor used in our experiments in section 4.*

## 2. Background and Related Work

The impact of the latent distribution on the performance of a generative model has been a significant topic of investigation. The choice of latent distribution has a significant impact on model performance in different types of architectures, including GANs (Karras et al., 2019; Ben-Yosef & Weinshall, 2018; Mukherjee et al., 2019; Arici & Celikyilmaz, 2016; Brock et al., 2019; Xiao et al., 2018; Luise et al., 2020) and flow matching models (Tong et al., 2023; Lee et al., 2026). Some general arguments have been put forward to explain why, for example, uncorrelated latent distributions are not well suited for capturing correlated features in complex data (Karras et al., 2019). However, developing a more general theoretical understanding of the relationship between the latent distribution and the data distribution has been challenging. To address this issue, (Hu et al., 2023) introduced a novel distance between these two spaces, which is minimized when the latent most effectively capitalizes on the capacity of the generator neural network. In our work, we extend this formalism to develop an understanding of the impact of the computational complexity of the latent distribution on model performance.

There have been several previous investigations of quantum latent distributions in generative models with a focus on GANs, in which empirical performance improvements were observed. However, much of this work focused on demonstrating feasibility, with a smaller emphasis on benchmarking or on developing theoretical foundations for this approach. In (Vakili et al., 2025) a quantum latent distribution is used within a generative algorithm to propose new molecules for potential medical applications, and in (Xiao et al., 2024) this approach is used for image reconstruction. In (Li et al., 2021; Kao et al., 2023) and (Wilson et al., 2021; Rudolph et al., 2022), small-scale quantum latent distributions in GANs are found to outperform a classical latent distribution on the QM9 and MNIST datasets. However, in these works the use of a trained quantum distribution compared to an untrained classical baseline makes it difficult to draw generalizable conclusions. (Jin & Merz Jr, 2025) empirically show an improvement in the BigGAN model (Brock et al., 2019) using a range of untrained quantum latent distributions, but do not benchmark against other classical latent distributions or provide theoretical foundations for this approach. We go beyond this prior work both by developing a theoretical understanding of the use of quantum distributions in these models, and by benchmarking against a range of classical distributions in an apples-to-apples experimental setting.

### 2.1. Quantum circuit sampling

Some classes of discrete distribution sampling tasks can be performed faster on near-term quantum computers than on classical computers (Hangleiter & Eisert, 2023). Two specific realizations of quantum computers that can perform such tasks are random circuit sampling (Movassagh, 2023; Arute et al., 2019) and boson sampling (Aaronson & Arkhipov, 2011; Zhong et al., 2020). Our work uses

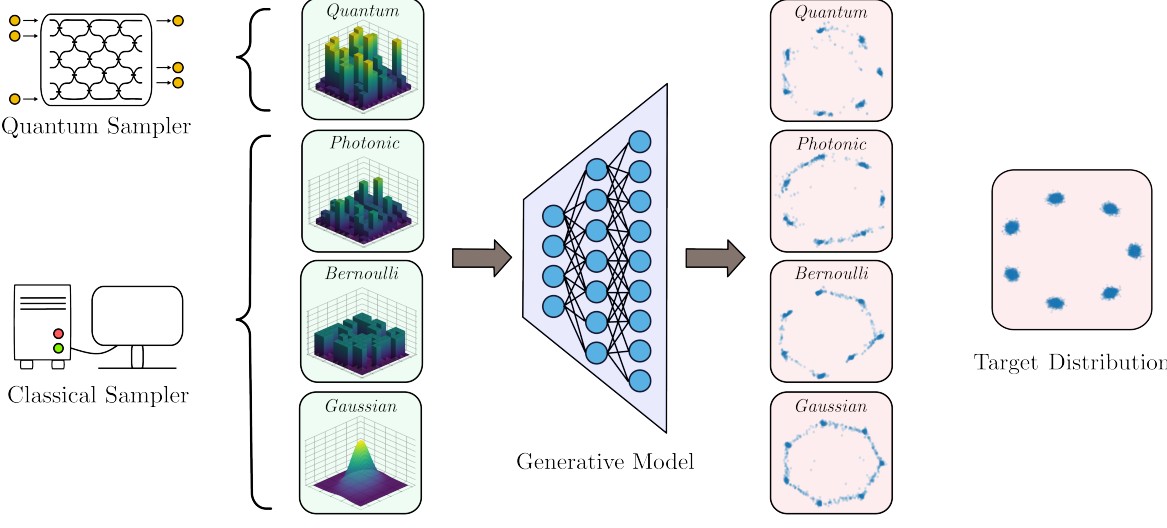

*Figure 1.* Our work provides a theoretical and empirical study of the use of non-classical distributions produced by quantum processors in the latent space of generative models. Even on a simple dataset such as the 2D mixture of Gaussians shown here, training a simple GAN with different latent distributions yields very different results. We compare four latent distributions representing a progression from the commonly used Gaussian distribution (bottom) to a quantum distribution produced by a photonic quantum processor (top). The main failure mode is a tendency to interpolate between different modes in the data, and the model with the quantum latent distribution is least affected. Further information on this experimental setting can be found in Appendix C. In this paper, we investigate this approach from a theoretical perspective, propose some intuition to explain these results, and conduct rigorous benchmarking experiments to compare quantum distributions to relevant classical distributions in an apples-to-apples setting.

photonic boson sampling systems, in which identical single photons are sent into a random interference circuit, creating an entangled state. Measurements of the output locations of these photons provide samples from a classically-intractable probability distribution.

The use of boson sampling distributions to investigate the impact of a quantum distribution in the latent space is motivated by several considerations. First, though they are not universal for quantum computation, they naturally produce highly non-uniform distributions (Aaronson & Arkhipov, 2013) that differ significantly from commonly used classical latent distributions. Moreover, photonic systems are subject to photon loss but not to decoherence. In other platforms, decoherence pushes the outputs of quantum circuits towards a classical uniform distribution (Wang et al., 2021), making it harder to observe a difference in performance (Jin & Merz Jr, 2025). These properties of photonic systems were leveraged in (Cimini et al., 2025; Yin et al., 2025) to perform some the largest-scale demonstrations of quantum machine learning to date.

## 3. Quantum distributions in generative models

In this section we introduce complexity classes for quantum and classical sampling tasks. We establish conditions under which a quantum latent distribution can lead to improved model performance compared to classical distributions. We

present theorems without proof, with complete proofs and rigorous definitions found in the appendix.

### 3.1. Preliminaries

**Sampling problems:** We consider the problem of drawing samples from a given probability distribution. We define $\mathcal{C}$ as the class of probability distributions that can be approximately sampled in $\text{Poly}(n, 1/\epsilon)$ time using classical computers. Here, $n$ represents the dimension of the samples to be generated, and $\epsilon$ the error within which we sample from this distribution. We also define $\mathcal{Q}$ as the class of distributions that can be sampled in $\text{Poly}(n, 1/\epsilon)$ with a quantum computer, but not with a classical computer. We provide rigorous definitions of these classes in the appendix.

In this section, we consider a family of neural network architectures $G$ mapping from $\mathbb{R}^{d_z}$ to $\mathbb{R}^d$, where $d_z$ is the latent space dimension and $d$ is the data space dimension with $d_z \leq d$. Similarly to (Hu et al., 2023), we assume that these architectures have bounded complexity according to some measure, such that the output can be efficiently produced from the input. Possible metrics for complexity can include width, depth, or Lipschitz constant of the network.

**GAN-induced distances:** Next, we consider a metric introduced in (Hu et al., 2023) to quantify the performance of a latent distribution for a generative task. For any distance between distributions $D(.,.)$ in the data space, we define a

generalized distance associated with neural network architectures in $G$ as

$$D^G(P_z, P_x) = \inf_{g \in G} D(P_{g(z)}, P_x) \qquad (1)$$

where $P_z$, $P_x$ denote the probability density of distributions defined over the latent space and the data space, respectively, and $P_{g(z)}$ is the pushforward distribution. This metric is particularly well suited for GAN architectures, since it can be viewed as a discriminator loss, and also introduces a notion of similarity between distributions in different dimensions. Informally, a latent distribution $P_z$ is best suited for generating a data distribution $P_x$ using generators within $G$ if it minimizes this distance.

### 3.2. Computational complexity of the latent distribution

Our first contribution is to formally establish the conditions under which a quantum latent distribution remains non-classical after being transformed by a neural network. We start with a simple observation about the complexity of classical pushforward distributions.

**Remark 1.** *Consider a latent distribution $P_z \in \mathcal{C}$ and a neural network $g \in G$ that is Lipschitz-continuous. Then the pushforward distribution $P_{g(z)} \in \mathcal{C}$.*

However, the pushforward distribution of a quantum distribution in $\mathcal{Q}$ is not necessarily in $\mathcal{Q}$. There are scenarios in which an initial quantum distribution is converted to a classical distribution, for example if the neural network produces constant outputs. Here, we establish sufficient conditions under which the pushforward distribution of a quantum distribution cannot be efficiently sampled from classically:

**Theorem 1.** *We consider a neural network $g \in G$ such that its inverse $g^{-1}$ exists, is efficiently classically implementable and is also Lipschitz continuous. Let $P_z$ be in $\mathcal{Q}$. Then the pushforward distribution $P_{g(z)}$ is not in $\mathcal{C}$.*

*Proof sketch*: With these assumptions on the generator, we can show that if the pushforward distribution is in $\mathcal{C}$, then by inverting $g$ the latent distribution can also be efficiently sampled and would therefore be in $\mathcal{C}$.

We can explicitly construct a deep neural network architecture that provably satisfies these assumptions. Consider a multi-layer perceptron mapping a small-dimensional latent space to a larger-dimensional data space through linear layers of increasing width with an invertible LeakyReLU activation. For such a model, the inputs can in general be efficiently reconstructed from the inputs by inverting a set of linear equations. This architecture satisfies the assumptions. In our benchmarking on the QM9 dataset, we adopt a model architecture that matches this example.

Even when a neural network does not strictly satisfy these

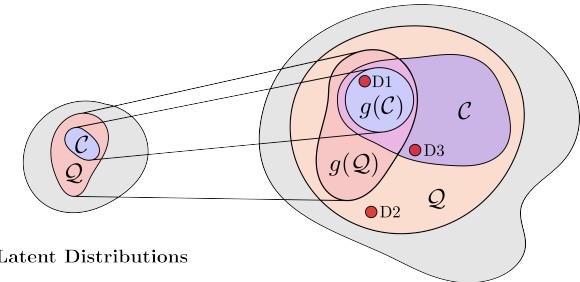

*Figure 2.* An illustration of the relation between the complexity of latent distributions and that of the pushforward distributions. $\mathcal{C}$ represents classical distributions, $\mathcal{Q}$ represents non-classical distributions, and $g(\mathcal{Q})$ and $g(\mathcal{C})$ are the pushforward distributions achieved using a class of neural networks $g$ with finite complexity. When the target dataset is in the pushforward distribution of $\mathcal{C}$ (D1), a quantum distribution is not expected to provide a benefit. However, when the target dataset is not in the pushforward distribution of $\mathcal{C}$ and can be either quantum (D2) or classical (D3) the pushforward distribution of a quantum latent distribution may be closer, leading to better performance.

sufficient conditions, Theorem 1 provides a basis for suggesting that many modern neural network architectures can map a non-classical distribution to another non-classical distribution. For example, though most neural networks are not analytically invertible, the "generator inversion" problem of finding a latent vector that produces a given output can be tractable in practice (Creswell & Bharath, 2018; Abdal et al., 2019). Moreover, a wide range of neural network architectures are Lipschitz-continuous (Virmaux & Scaman, 2018; Castin et al., 2024).

### 3.3. Impact on model performance

Here, we discuss the implications to model performance, defined in terms of GAN-induced distance, through the use of a quantum latent distribution. First, we show that at least some data distributions can be better approximated using a quantum latent distribution:

**Corollary 1.** *We consider a neural network $g$ satisfying the conditions of Theorem 1, and a quantum latent distribution $P_{z_\mathcal{Q}} \in \mathcal{Q}$. Using the Wasserstein distance as our metric, we have both $D^G(P_{z_\mathcal{Q}}, P_{g(z_\mathcal{Q})}) = 0$ by definition, and for any classical latent distribution $P_{z_\mathcal{C}} \in \mathcal{C}$ there exists $\epsilon > 0$ such that $D^G(P_{z_\mathcal{C}}, P_{g(z_\mathcal{Q})}) > \epsilon$.*

This shows that the class of pushforward distributions achieved by a quantum latent distribution under certain conditions cannot be approximated with any choice of classical latent distribution.

We can now reason in terms of performance on different types of datasets. For a target data distribution that is in $\mathcal{C}$, we can distinguish between two cases illustrated in figure 2. First, if this distribution is in the pushforward distribution of classical latents that are also in $\mathcal{C}$, then there is no theoret-

ical advantage to using a quantum latent distribution. In the second case, the target distribution is not in the pushforward distribution of classical latents. Since we consider generators with finite capacity and latent spaces that are smaller than the data space, this may be a common scenario. In that case, some achievable quantum pushforward distributions will be closer than any achievable classical pushforward distributions in terms of GAN-induced distance. As such, a quantum latent distribution may lead to an advantage even for a classical data distribution.

### 3.4. Achieving an advantage in practice

The results above establish a theoretical basis for the use of quantum distributions in generative models, but the conditions described above for achieving a performance advantage over classical distributions are still abstract. In practice, the benefit of a quantum latent distribution on real datasets may be best understood as that of a structured prior whose statistical properties are well matched to the data, rather than as a strict computational separation from classical alternatives. Here, we propose two practical mechanisms through which quantum distributions can lead to improved performance.

First, a consequence of multi-particle entanglement is that quantum distributions cannot be factorized into independent factors of variation. This prevents a model from learning a factorized representation of the data. Though factorizable representations are often viewed positively since they are more interpretable, empirical evidence for other benefits are mixed (Locatello et al., 2019; Montero et al., 2022). Factorized representations can perform poorly on multimodal datasets without additional modifications (Tsai et al., 2019), and non-factorizable determinantal point processes have been integrated into GANs (Elfeki et al., 2019) and flow matching (Morshed & Boddeti, 2025) to improve output diversity. We hypothesize that using a non-factorizable distribution can in some cases improve learning by biasing the training process towards more complex models for the data, avoiding oversimplified models that focus on uncorrelated features. Quantum distributions exhibit a strong form of non-factorizability: unlike classically tractable non-factorized distributions, which can still be transformed into factorized distributions when their probabilities are efficiently computable with methods such as cumulative transforms or rejection sampling, no such efficient transformation is possible in general for quantum distributions.

Second, quantum distributions are highly non-uniform, with strong correlations at multiple orders (Aaronson & Arkhipov, 2013; Phillips et al., 2019). We anticipate that this can provide a beneficial inductive bias for datasets that have similar properties. For example, a quantum latent distribution can be expected to provide such a bias for a dataset

arising from a fundamental physical process that is better described by quantum mechanics than by classical mechanics. However, though non-uniformity and strong correlations are features of quantum distributions that may provide a benefit, we note that these are not uniquely quantum features and other classical distributions share similar properties. To help assess whether the specific statistics arising from quantum mechanics can be beneficial, our experiments in the next section compare the performance of a photonic quantum distribution to that of a photonic distribution where quantum interference effects are switched off.

To test these intuitions, we trained GANs with two hidden layers and different latent distributions (described in the next section) on a 2D mixture of Gaussians dataset. Though simple, this dataset is highly non-uniform and multimodal, making it a priori a good match for the properties of quantum latent distributions. Our results are shown in figure 1 and support these intuitions, where we observe that the quantum distribution produces the model that most closely matches the data. This is in line with prior work from (Luise et al., 2020), where up to 7 layers were required to successfully map a Gaussian latent distribution to a mixture of Gaussians.

## 4. Benchmarking GANs with quantum latent distributions

In this section, we empirically investigate whether the use of a quantum latent distribution can lead to a performance improvement over relevant classical distributions. We consider generative adversarial networks (GANs), where the use of different latent distributions is well-studied. Though GANs are no longer state of the art for several types of data, they provide a straightforward mapping from latent distribution to data distribution that facilitates apples-to-apples comparisons between latent distributions.

We perform two experiments: one on a small-scale synthetic dataset produced by a quantum process, and one on the QM9 quantum chemistry dataset. Based on the intuition developed in the previous section, these datasets have properties that make them a good fit for testing a quantum latent distribution. The synthetic dataset is by construction sampled from a distribution in $\mathcal{Q}$. For QM9, the chemical properties of molecules arise from quantum mechanics. Though there is no formal argument that the distribution of molecules in QM9 belongs to any specific complexity class, this serves as a motivating heuristic for our choice of this dataset.

Our objective is to provide a controlled experimental comparison between different latent distributions. For both experiments, we compare different types of latent distributions that produce samples $z$ of length $d_z = L$. We select one quantum distribution as well as three classical latent distributions:

- **Quantum**: arising from indistinguishable photons sent into a size-$L$ interferometer

- **Photonic**: arising from distinguishable photons sent into a size-$L$ interferometer

- **Bernoulli**: random uniform bit strings (discrete), $z \in \{0,1\}^L$

- **Gaussian**: the commonly used Gaussian distribution (continuous), $z \in \mathbb{R}^L \sim \mathcal{N}(0, I)$

Samples from both the quantum and photonic distributions are vectors of photon counts across the $L$ output channels of the interferometer, with N input photons distributed across these channels such that each sample sums to $N$. For example, with 3 photons in 6 channels, possible samples include $[1, 1, 0, 1, 0, 0]$ or $[1, 0, 0, 2, 0, 0]$. The two distributions differ in how these counts are produced. In the photonic case, each input photon is independently routed to an output channel, with probabilities determined by how its input channel is coupled to the outputs through the interferometer; the overall sample is then the sum of these independent single-photon outcomes. Though self-interference of each individual photon is still present in this case, each photon is effectively independent. The quantum distribution, by contrast, is shaped by more complex multi-photon interference and cannot in general be decomposed into independent single-photon contributions.

The comparison between the quantum and photonic distributions isolates multi-photon quantum interference as the single factor that differs between the two, allowing us to assess its specific contribution to performance. The comparisons against Gaussian and Bernoulli distributions provide standard baselines to show that the quantum distribution can be competitive against common choices for a latent distribution. Other classical distributions could in principle be considered, but introduce confounding factors that make an apples-to-apples comparison more difficult. For example, comparing to trained priors such as normalizing flows would introduce additional training complexity that makes it hard to separate the contribution of the latent structure from that of the additional training.

To ensure an apples-to-apples comparison, for all experiments our models differ only through the nature of the latent distribution. Moreover, both the quantum and photonic distributions arise from randomly initialized interference circuits. To ensure that performance differences arise from general properties of these distributions and not from specific circuit realizations, these circuits are re-sampled for every seed (with an exception for the largest-sized latent spaces, see appendix for details). We note that these circuits could additionally be trained (Facelli et al., 2024), although training methods for quantum circuits often scale poorly

(McClean et al., 2018). We do not train them in this work since this could lead to an unfair advantage in benchmarking against the static Gaussian and Bernoulli distributions.

### 4.1. Experiment on synthetic datasets

For our small-scale experiment, we consider two synthetic datasets: a "Quantum dataset" produced by simulating (Clifford & Clifford, 2018) 8 identical photons interfering in a 16-channel random optical circuit, and a "Bernoulli dataset" produced by a 16-dimensional Bernoulli distribution. The optical circuits used for the latent and data distributions are independently sampled, preventing a trivial identity mapping from the latent space to the data space. Since the two target distributions are discrete, to quantify the performance of a trained model we use the average distance between the outputs of the generator and their nearest integers. A smaller distance indicates that the generator has been more successful in learning the discrete nature of the distribution.

*Table 1.* L1 distances between the numbers generated by the GANs and their nearest integers. Each row corresponds to a different latent distribution, and each column corresponds to a different dataset. The error bounds represent the uncertainty of the mean, estimated over 12 runs.

|  | Quantum dataset | Bernoulli dataset |
|---|---|---|
| Quantum | **0.036 ± 0.001** | 0.015 ± 0.002 |
| Photonic | 0.041 ± 0.002 | 0.017 ± 0.002 |
| Bernoulli | 0.065 ± 0.001 | 0.020 ± 0.013 |
| Gaussian | 0.061 ± 0.001 | **0.012 ± 0.002** |

Our results comparing the performance of the trained models are shown in Table 1. We first observe that the distances and differences between latents are smaller on the classical dataset. Though the two columns are not exactly comparable since the quantum and Bernoulli datasets have different supports and the absolute distances depend in part on the support, this does suggest that the classical dataset is easier to learn. On the harder quantum dataset, we observe significant differences between the latent distributions, with the quantum latent distribution achieving the best results. We find that the quantum latent distribution outperforms the latent distribution that uses distinguishable photons. This implies that the statistics arising from quantum interference between photons are a useful resource that the model used to achieve better performance on the quantum dataset. These results validate the expectation that a quantum distribution can be a suitable latent distribution when the data also arises from a quantum process.

### 4.2. Experiments on the QM9 dataset

Next, we consider the QM9 quantum chemistry dataset (Ramakrishnan et al., 2014). Generating chemical structures

with specific properties is a challenging problem, partly because the underlying dynamics are described by quantum mechanical principles. The QM9 dataset is a reasonably small-size dataset that is amenable to extensive benchmarking, and which has been previously studied within the context of GANs (De Cao & Kipf, 2018; Guimaraes et al., 2017; Li & Yamanishi, 2024). We use a GAN architecture based on MolGAN (De Cao & Kipf, 2018), with some improvements described in the appendix. The generator is a feedforward neural network which is based on the requirements of Theorem 1: all layers are non-decreasing, and we use a LeakyReLU activation function. The discriminator is a relational graph convolutional network.

We perform a wide range of experiments, with 20 random seeds per experiment, in which we test different combinations and sizes of latent distributions. These experiments allow us to determine how different latent distributions can affect performance. To quantify performance, we use the Frechet Chemical Distance (FCD) (Preuer et al., 2018), the number of valid and unique molecules generated by the model from 10k generation attempts (# Valid), and the number of molecules among this set which are novel and not in the training set (# Novel). A successful model has low FCD and generates high numbers of valid and unique molecules that are not in the training set. We note that all these metrics correlate with the diversity of the generated data.

### 4.2.1. TYPE AND SIZE OF THE LATENT DISTRIBUTION

We first compare results over different latent distributions and different latent space sizes: 16, 32 and 48. For the distinguishable and indistinguishable photon distributions, we consider unstructured optical circuits, with transfer matrices described by Haar-random unitary matrices, and with half as many input photons as optical channels. For example, the size-16 latent is produced by 8 photons in 16 channels. We use the algorithm in (Clifford & Clifford, 2018) to simulate these systems.

Our results are shown in table 2, with additional results such as training curves in the appendix. We find that, for all 3 latent space sizes, the quantum distribution achieves the best results. The largest performance improvement occurs for the size-16 latent space, where the quantum distribution outperforms all other latent distributions on all metrics. Though the gap narrows for larger latent spaces, and overall performance decreases, the size-48 quantum latent distribution still leads to a higher number of valid and unique generated molecules than the classical baselines. Moreover, as shown by the improvement over distinguishable photons, the statistics arising from quantum interference is a factor that improves performance. These results indicate that a quantum latent distribution can outperform other relevant classical latent distributions on chemistry data.

*Table 2.* Metrics (± standard error over 20 seeds) for different latents with different dimensions

| Latent Type | FCD ↓ | # Valid ↑ | # Novel ↑ |
|---|---|---|---|
| $d_z = 16$ | | | |
| Quantum | **1.160 ± 0.06** | **2522 ± 65** | **1331 ± 37** |
| Photonic | 1.333 ± 0.07 | 1954 ± 103 | 1067 ± 54 |
| Bernoulli | 1.822 ± 0.09 | 1244 ± 102 | 702 ± 56 |
| Gaussian | 1.529 ± 0.08 | 1814 ± 115 | 1017 ± 64 |
| $d_z = 32$ | | | |
| Quantum | 1.536 ± 0.08 | **1791 ± 106** | 951 ± 37 |
| Photonic | 1.533 ± 0.06 | 1633 ± 81 | 919 ± 45 |
| Bernoulli | 1.646 ± 0.09 | 1391 ± 77 | 790 ± 44 |
| Gaussian | 1.823 ± 0.07 | 1320 ± 53 | 768 ± 35 |
| $d_z = 48$ | | | |
| Quantum | 1.696 ± 0.08 | **1528 ± 65** | 856 ± 40 |
| Photonic | 1.713 ± 0.06 | 1307 ± 77 | 746 ± 43 |
| Bernoulli | 1.671 ± 0.08 | 1456 ± 82 | 816 ± 48 |
| Gaussian | 1.781 ± 0.07 | 1352 ± 58 | 766 ± 30 |

### 4.2.2. TYPE OF QUANTUM CIRCUIT

To understand whether the results presented in the previous section are specific to Haar-random optical circuits or are due to more general properties of the statistics of photon interference, we also run experiments with different types of optical circuits. We simulate two realizations of boson sampling circuits based on optical delay lines, which have been proposed and demonstrated as an experimentally feasible route to large-scale boson sampling (Madsen et al., 2022; Deshpande et al., 2022; Novák et al., 2025). These realizations respectively have two delay lines of the same length (a "1-1" configuration), and three delay lines in a "1-3-9" configuration where each line delays each photon by three times the previous line.

*Table 3.* Metrics (± standard error over 20 seeds) for different size-16 latents using different optical circuits

| Latent Type | FCD ↓ | # Valid ↑ | # Novel ↑ |
|---|---|---|---|
| 1-1 configuration | | | |
| Quantum | 1.245 ± 0.06 | **1375 ± 72** | **692 ± 34** |
| Photonic | 1.291 ± 0.07 | 1182 ± 96 | 580 ± 47 |
| 1-3-9 configuration | | | |
| Quantum | **1.187 ± 0.06** | **1990 ± 106** | **1063 ± 58** |
| Photonic | 1.376 ± 0.09 | 1459 ± 100 | 775 ± 53 |

Our results are shown in Table 3. We find that for each type of optical circuit, the quantum distribution outperforms the classical distribution arising from distinguishable photons.

This shows that for this model and dataset the statistics arising from quantum interference confer an advantage that does not depend on the specifics of the circuit architecture. We also find that the circuit architecture has an impact on the overall performance of the latent distribution, with the Haar-random circuit from table 2 performing best overall for size-16 latents. This suggests that developing methods for circuit architecture search or joint training of the circuit with the neural network could lead to improved results.

### 4.2.3. USING A REAL QUANTUM PROCESSOR

Real boson sampling systems are subject to a range of imperfections such as optical loss, limited photon number resolution, and imperfect indistinguishability. To assess the extent to which our previous results are transferrable to a real quantum processor, we performed experiments on a commercially available boson sampling system. This system can interfere 16 photons in 32 channels and its optical network consists of optical delay lines in the "1-1" configuration. More information about this system can be found in the appendix. We collected half a million samples from this system with a single circuit realization, and compared the results to comparable size-32 latent distributions.

*Table 4.* Metrics (± standard error over 20 seeds) for a real quantum device and simulated size-32 latents

| Latent Type | FCD ↓ | # Valid ↑ | # Novel ↑ |
|---|---|---|---|
| Quantum (real) | **1.410 ± 0.07** | **1962 ± 75** | **1084 ± 45** |
| Quantum (sim) | **1.414 ± 0.07** | **1903 ± 66** | **1073 ± 36** |
| Photonic | 1.560 ± 0.09 | 1638 ± 102 | 900 ± 50 |
| Bernoulli | 1.646 ± 0.09 | 1391 ± 77 | 790 ± 44 |
| Gaussian | 1.823 ± 0.07 | 1320 ± 53 | 768 ± 35 |

As shown in Table 4, the results from the real quantum processor closely match the simulated system despite experimental imperfections. Both quantum distributions also outperform the classical distributions. These results indicate that a performance advantage achieved with a simulated quantum system can provide a good indication of the performance of a real quantum processor, which can be helpful for extrapolating the performance of small-scale simulable distributions to larger-scale, non-simulable distributions.

## 5. Discussion

While the magnitude of the improvement varies across latent dimensionalities and experimental settings, our empirical study shows that quantum latent distributions can match or exceed the performance of relevant classical baselines on generative chemistry tasks. Notably, quantum interference effects appear to be a meaningful source of inductive bias for the QM9 dataset. These results provide experimental

support for the theoretical arguments of Section 3: that quantum latent distributions may be beneficial in settings where data distributions are strongly correlated, multimodal, or rooted in quantum mechanical processes.

These experiments were limited to size-48 latents due to the exponentially increasing complexity of simulating a quantum processor. Generating 1 million size-48 samples with the algorithm in (Clifford & Clifford, 2018) took about 3.75 hours on 100 AMD EPYC™ 7003 processors, which may be close to a regime where a near-term quantum processor would run faster. The persistence of a small advantage at size 48 on the number of valid, unique and valid molecules suggests that an advantage over the classical distributions may still be achievable for a quantum processor of a size that is not practically simulable. Though for these experiments a smaller latent space yields improved results, for other models and datasets larger-scale latents have been shown to work best (Brock et al., 2019), which would motivate further exploration of this approach for larger-scale quantum distributions.

Though our work has focused on GANs, the theoretical understanding of quantum latent distributions developed in this work can also be extended to other types of models. Though some models such as diffusion models often make specific Gaussianity assumptions about the latent space, other models can flexibly map any two distributions only from samples without these assumptions. These include several modern flow matching models (Albergo et al., 2025), which can be highly sensitive to the choice of latent distribution (Lee et al., 2026). In the appendix we demonstrate implementations of diffusion and flow matching models that use a quantum latent distribution. We emphasize that these are compatibility demonstrations intended to show that quantum latent distributions can be integrated into these architectures, rather than performance validations. A thorough benchmarking of quantum latent distributions in these models is left to future work.

We also note that the benefits of a quantum latent distribution are not universal. In Appendix H, we also report negative results on QM9 under a different training regime and on the CIFAR-10 dataset with a StyleGAN architecture (Karras et al., 2019). These results are consistent with our theoretical framework, which predicts that an advantage should depend on the interplay between the latent distribution, the model architecture, the training procedure, and the structure of the data. Identifying more precisely which combinations of these factors give rise to a improved performance with a quantum latent distribution is an important direction for future work.

## 5.1. Classical alternatives

Though our comparison between distinguishable and indistinguishable photons show that the statistics arising from quantum interference can improve performance, a relevant question is whether other classical distributions may be able to reproduce similar statistics without a quantum processor. In particular, classical methods for simulating photonic quantum processors in the presence of noise have been developed (Villalonga et al., 2021; Oh et al., 2024). These methods may be able to achieve a similar performance improvement, however they are likely to be impractical for machine learning applications. The approach of (Villalonga et al., 2021) reproduces the low-order statistics of a boson sampling distribution with a quadratic overhead compared to a linear overhead at most with a real physical system. The alternative approaches of (Oh et al., 2024; Dodd et al., 2025) require a significant amount of time to initially set up the simulation. Neither approach is thus likely to run faster than a real quantum processor. In a more advanced setting where the latent distribution were to be trained jointly with the neural network, for example using the method in (Facelli et al., 2024), slower speeds are likely to make these existing classical approximation methods unsuitable.

An alternative to approximated quantum distributions to achieve a performance improvement would be to use entirely different classes of classical distributions. More complex classical distributions could be used, for example trained distributions using an autoencoder at the expense of additional training (Hu et al., 2023). However, testing a wide range of alternative classical distributions is impractical and computationally expensive. On the other hand, quantum computers are progressively becoming more widely available, in both datacenter environments and in the cloud (Devitt, 2016; Madsen et al., 2022; Beck et al., 2024; Slysz et al., 2025). While cost, complexity and availability remain real practical considerations, these are expected to decrease as the technology matures. In the settings identified in this work in which a quantum latent distribution could be expected to yield good results, a quantum computer may be an effective use of resources.

## 6. Conclusion

In this work, we have developed an understanding of how quantum latent distributions can have an impact on the outputs of generative models. While recent experiments had already observed a performance advantage on some datasets, here we established theoretical foundations for this approach, provided intuition as to the possible causes of this improved performance, and presented rigorous benchmarks that validate these intuitions.

These results motivate further exploration of quantum latent distributions in modern generative architectures. More broadly, our results highlight that the latent space is an important component of inductive bias in a model, and that quantum distributions provide a distinct and practically accessible family of biases. Exploring this space in larger and more expressive generative models is a compelling direction for future work.

## Impact Statement

This paper presents work whose goal is to advance the field of Machine Learning. Part of our work concerns generative modelling of molecules, which carries dual-use risks: while such models are primarily intended to support beneficial applications such as drug discovery, similar techniques could in principle be misused to propose harmful chemical structures. Our experiments use a small public dataset of small organic molecules and are methodological in nature, but we encourage practitioners building on this line of research at larger scales to consider appropriate safeguards.

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

# A. Complexity Proofs

Here we give precise definitions and complete proofs for the results stated in Section 3. We begin by rigorously defining the complexity classes for drawing samples from a given probability distribution.

In (Aaronson & Arkhipov, 2011), SampP is defined as the class of sampling problems that can be approximately solved by a probabilistic polynomial-time classical algorithm, and SampBQP is the class of sampling problems that can be approximately solved in polynomial-time by a quantum algorithm. Both of these complexity classes are defined in terms of binary variables. Though this does account for binary storage methods of finite-precision continuous values, this is not a natural setting when considering latent distributions for generative models.

Within this paper, we consider distributions over continuous spaces, which motivates extending these complexity classes to include continuous notions of approximate sampling using metrics such as the Wasserstein metric. For the classes of quantum distributions we consider, we similarly wish to step away from the language of binary variables in order to facilitate less constrained descriptions of sampling distributions such as boson sampling in which measurement results are discrete (the number of photons in an optical channel) but not binary. As such, we introduce two new complexity classes we will use throughout this section.

**Definition:** *Efficient Approximate Continuous Classical Sampling*

We define, with $\mathcal{C}$, the space of sampling problems of the form $\mathcal{S} = (P_{z_n})_{n \in \mathbb{N}}$ (where each $P_{z_n}$ is a probability distribution over $\mathbb{R}^n$) such that we can sample from distributions $(\hat{P}_{z_n})_{n \in \mathbb{N}}$ with $W(P_{z_n}, \hat{P}_{z_n}) < \epsilon$ using a classical algorithm in Poly$(n, 1/\epsilon)$ time. Here, $W$ denotes the Wasserstein metric

**Definition:** *Efficient Approximate Discrete Quantum Sampling*

We define, with $\mathcal{Q}$, the space of sampling problems of the form $\mathcal{S} = (P_{z_n})_{n \in \mathbb{N}}$ (where each $P_{z_n}$ is a probability distribution over $\mathbb{N}^n$) such that we can sample from a distribution $\hat{P}_{z_n}$ with $\|P_{z_n} - \hat{P}_{z_n}\| < \epsilon$ using a quantum circuit in Poly$(n, 1/\epsilon)$ time, and there exists no classical algorithm that can do the same. Here, $\| \cdot \|$ denotes total variation distance.

We note that, in the scenario where each entry in $\mathbb{N}$ is restricted by a given upper bound, this class is exactly equivalent to SampBQP/SampP. Boson sampling from $n$-mode circuits is a task thought to be in $\mathcal{Q}$ (Aaronson & Arkhipov, 2013).

We use different metrics for the proximity of probability distributions for each class in order to capture two distinct notions of 'closeness' when discussing discrete and continuous sampling tasks. We note that the SampP class is not a strict subset of the class $\mathcal{C}$ as the latter uses Wasserstein distance to define the closeness of distributions rather than total variation distance. For our purposes of considering continuous classical latent distributions and data distributions, we will use the Wasserstein metric.

Next, we describe the setting that we use for the generative models we study throughout this section.

**Generative Model Setting:** We denote our collection of latent distributions as $(P_{z_n})_{n \in \mathbb{N}}$, where $P_{z_n}$ is a distribution over $\mathbb{R}^n$. We will consider collections of neural network architectures $(g_n)_{n \in \mathbb{N}}$ where $g_n$ takes as input a vector of length $n$. In this section, we will assume these architectures are all efficiently classically implementable in the sense that there exists a Poly$(n)$ classical algorithm that applies the collection of functions. We will generally use $(P_{g_n(z_n)})_{n \in \mathbb{N}}$ to denote the pushforward distributions, where:

$$g_n(Z) \sim P_{g_n(z_n)} \quad \text{where} \quad Z \sim P_{z_n}$$

**Remark 1**: If the collection of latent distributions $(P_{z_n})_{n \in \mathbb{N}} \in \mathcal{C}$, and the neural networks $(g_n)_{n \in \mathbb{N}}$ are Lipschitz with constant $c$, then the pushforward distributions $(P_{g_n(z_n)})_{n \in \mathbb{N}} \in \mathcal{C}$.

**Proof**: Let $(\hat{P}_{z_n})_{n \in \mathbb{N}}$ be the collection of efficiently classically sampleable approximations guaranteed to exist by $(P_{z_n})_{n \in \mathbb{N}} \in \mathcal{C}$. Let $\hat{P}_{g_n(z_n)}$ denote the distribution sampled from by applying $g_n$ to values sampled according to $\hat{P}_{z_n}$. We

note:

$$
\begin{aligned}
W(P_{g_n(z_n)}, \hat{P}_{g_n(z_n)}) &= \inf_{\gamma \in \Gamma(P_{g_n(z_n)}, \hat{P}_{g_n(z_n)})} \mathbb{E}_{(X,Y)\sim\gamma}|X - Y| \\
&= \inf_{\gamma \in \Gamma(P_{z_n}, \hat{P}_{z_n})} \mathbb{E}_{(X,Y)\sim\gamma}|g_n(X) - g_n(Y)| \\
&\leq c \cdot \inf_{\gamma \in \Gamma(P_{z_n}, \hat{P}_{z_n})} \mathbb{E}_{(X,Y)\sim\gamma}|X - Y| \\
&= cW(P_{z_n}, \hat{P}_{z_n}) \\
&< c\epsilon
\end{aligned}
$$

As we can apply $(g_n)$ in Poly$(n)$ time, we can sample from $(\hat{P}_{g_n(z_n)})_{n\in\mathbb{N}}$ in Poly$(n, 1/\epsilon)$ time. Thus, $(P_{g_n(z_n)})_{n\in\mathbb{N}} \in \mathcal{C}$.
$\square$

**Theorem 1**: Let the neural networks $(g_n)_{n\in\mathbb{N}}$ be invertible such that there is a classical algorithm that applies $g_n^{-1}$ in Poly$(n)$ time and further, each $g_n^{-1}$ is Lipschitz continuous with constant $c$. Let $(P_{z_n})_{n\in\mathbb{N}}$ be in $\mathcal{Q}$. Then, sampling from the collection of pushforward distributions $(P_{g_n(z_n)})_{n\in\mathbb{N}}$ is not in $\mathcal{C}$.

**Proof**: For a contradiction, we assume that the sampling task defined by $(P_{g_n(z_n)})_{n\in\mathbb{N}}$ is in $\mathcal{C}$. Let $(\hat{P}_{g_n(z_n)})_{n\in\mathbb{N}}$ denote the classically efficiently sampleable approximation guaranteed to exist by the definition of $\mathcal{C}$. Let $\hat{P}_{z_n}$ denote the distribution sampled by applying $g_n^{-1}$ to values sampled according to $\hat{P}_{g_n(z_n)}$. We claim that $W(P_{z_n}, \hat{P}_{z_n}) < c\epsilon$. Indeed,

$$
\begin{aligned}
W(P_{z_n}, \hat{P}_{z_n}) &= \inf_{\gamma \in \Gamma(P_{z_n}, \hat{P}_{z_n})} \mathbb{E}_{(X,Y)\sim\gamma}|X - Y| \\
&\leq \inf_{\gamma \in \Gamma(P_{z_n}, \hat{P}_{z_n})} \mathbb{E}_{(X,Y)\sim\gamma} c \cdot |g_n(X) - g_n(Y)| \\
&= c \cdot \inf_{\gamma \in \Gamma(P_{g_n(z_n)}, \hat{P}_{g_n(z_n)})} \mathbb{E}_{(X',Y')\sim\gamma}|X' - Y'| \\
&= cW(P_{g_n(z_n)}, \hat{P}_{g_n(z_n)}) \\
&< c\epsilon
\end{aligned}
$$

Thus, $\exists$ a collection of couplings $\gamma_n^* \in \Gamma(P_{z_n}, \hat{P}_{g_n(z_n)})$ such that, $\forall n$:

$$
\mathbb{E}_{(X,Y)\sim\gamma_n^*}|X - Y| \leq c\epsilon \quad \Rightarrow \quad \mathbb{P}_{(X,Y)\sim\gamma_n^*}(|X - Y| \geq 0.5) \leq 2c\epsilon
$$

We note that, if $X \in \mathbb{R}^n, Y \in \mathbb{N}^n$ and $|X - Y| < 1/2$, then rounding every entry of $X$ to the nearest integer returns $Y$. Thus, the above result shows that sampling from $\hat{P}_{g_n(z_n)}$ and then rounding each entry to the nearest integer, samples from a distribution within $2c\epsilon$ total variation distance of $P_{z_n}$. Sampling from $(P_{g_n(z_n)})_{n\in\mathbb{N}}$ is in $\mathcal{C}$, applying $g_n^{-1}$ is done in Poly$(n)$ time and rounding decimal values to integers can be done in constant time so we can sample from this distribution classically in Poly$(n, 1/\epsilon)$ time. As these distributions are within $2c\epsilon$ total variation distance of $P_{z_n}$, the task of sampling from $(P_{z_n})_{n\in\mathbb{N}}$ within $\epsilon$ TVD can be done classically in Poly$(n, 1/\epsilon)$ time.[1] This contradicts $(P_{z_n})_{n\in\mathbb{N}} \in \mathcal{Q}$. We conclude the claimed result. $\square$

Recall that, in Section 3, we define GAN-induced distances. We will prove the following result with this language. First, we define $(G_{n,\epsilon})_{n\in\mathbb{N}, \epsilon\in\mathbb{R}_+}$ to be collections of neural network architectures, with the condition that there exists a Poly$(n, 1/\epsilon)$ time classical algorithm capable of implementing all functions in $G_{n,\epsilon}$. We fix $D$ to be the Wasserstein metric.

**Corollary 1**: Let us be in the same setting as described in Theorem 1. Let $(\hat{P}_{z_n})_{n\in\mathbb{N}} \in \mathcal{C}$ be a collection of latent distributions we can sample from exactly in Poly$(n)$ time. Let $(\hat{P}_{g_{n,\epsilon}(z_n)})_{n\in\mathbb{N}, \epsilon\in\mathbb{R}}$ be the resulting collection of pushforward distributions. There exists $\epsilon \in \mathbb{R}, n \in \mathbb{N}$ such that:

$$
D^{G_{n,\epsilon}}(\hat{P}_{z_n}, P_{g_n(z_n)}) \geq \epsilon
$$

whereas, by definition,

$$
D^{g_n}(P_{z_n}, P_{g_n(z_n)}) = 0 \quad \forall n
$$

---

[1]The TVD error we achieve is a constant multiple of $\epsilon$. As such we reach the same conclusions on complexity since Poly$(n, 1/c\epsilon) \equiv$ Poly$(n, 1/\epsilon)$ for some constant $c$

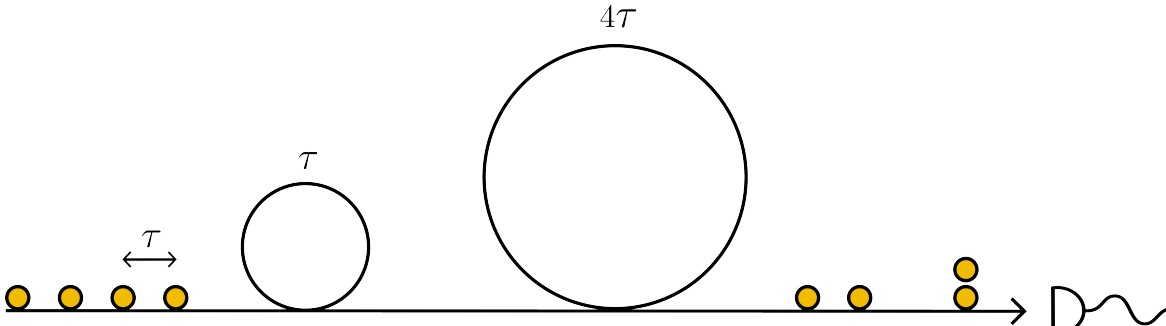

*Figure 3*. Illustration of a loop-based boson sampling system. Here, four sequential single photons, separated by time $\tau$, are sent into a system consisting of two sequential loops with lengths such that each photon that enters the loop can interfere with a subsequent photon. Sequences of loops of different lengths ensures that a photon can interfere with a photon several time-steps behind.

**Proof**: To work towards a contradiction, we assume that $D^{G_{n,\epsilon}}(\hat{P}_{z_n}, P_{g_n(z_n)}) < \epsilon$ holds $\forall n, \epsilon$. We then pick the functions $g_{n,\epsilon} \in G_{n,\epsilon}$ such that $W(\hat{P}_{g_{n,\epsilon}(z_n)}, P_{g_n(z_n)}) < \epsilon$. As we can sample from $(\hat{P}_{g_{n,\epsilon}(z_n)})$ in Poly$(n, 1/\epsilon)$ time, $(P_{g_n(z_n)})_{n \in \mathbb{N}} \in \mathcal{C}$. This contradicts Theorem 1. $\square$

Precisely, the result demonstrated is that, no matter the choice of latent distribution and efficiently implementable neural network architecture, it is not possible to efficiently classically simulate the performance of a specific set of GAN architectures whose latent distributions are generated by quantum circuits.

## B. Boson sampling systems

### B.1. Boson sampling theory

Boson sampling is a non-universal model of quantum computation proposed by (Aaronson & Arkhipov, 2011), in which identical photons are sent into an interference circuit, and a measurement is performed to determine where the photons left the circuit. A lossless interference circuit with $N$ channels is described by an $N \times N$ unitary matrix $U$, which can be physically implemented by a wide range of physical devices such as integrated chips (Spring et al., 2013) or optical fiber loops (Madsen et al., 2022). If $k$ identical photons are sent into the first $k$ channels of an interference device, then the probability of measuring an output configuration where $t_i$ photons were found in output channel $i$ is

$$p(t_1, ..., t_N) = \frac{|\text{Perm}(U_{k,T})|^2}{t_1! ... t_N!}$$

where Perm indicates the matrix permanent and $U_{k,T}$ is the submatrix of $U$ obtained by taking the first $k$ columns of $U$ and repeating each row $i$ $t_i$ times (if $t_i = 0$ then row $i$ is not in $U_{k,T}$). The permanent of an $n$ by $n$ matrix $M$ with elements $x_{i,j}$ is defined as

$$\text{Perm}(M) = \sum_{\sigma \in S_n} \prod_{i=1}^{n} x_{i,\sigma_i}$$

where $S_n$ is the set of all permutations of the numbers 1 to $n$. Although this expression is similar to that of a matrix determinant, unlike a determinant calculating the permanent of a matrix is in general a computationally hard problem. Indeed, if $M$ is an arbitrary real or complex matrix then this problem is #P-hard. Since calculating a permanent is hard, (Aaronson & Arkhipov, 2011) show that the problem of sampling from the output distribution of a boson sampler, even approximately, is also hard. The classical complexity of simulating a boson sampler increases exponentially with the number of photons, such that current classical computers cannot simulate more than a few tens of photons (Clifford & Clifford, 2018).

### B.2. Boson samplers with delay lines

Boson samplers based on optical delay lines have been proposed and demonstrated as an experimentally feasible route to large-scale boson sampling (Madsen et al., 2022; Deshpande et al., 2022; Novák et al., 2025). In these systems, illustrated in figure 3, a single photon source is used to produce a sequence of identical single photons, which then interfere with

each other in a sequence of optical delay lines (or "loops"). The resulting superposition state is measured by a single photon number resolving detector. The main advantage of this approach is that only a single photon source and detector are required, even though many photons and time-bins may be involved. Moreover, only three loops of different lengths are required to reach the classically hard to simulate regime (Deshpande et al., 2022; Novák et al., 2025). Furthermore, in (Madsen et al., 2022) and in the ORCA Computing PT-2, the coupling parameter between each loop and each input/output is re-programmable between each time step using fast electro-optic modulators. This opens the door to the possibility of jointly training these parameters with the parameters of a neural network.

### B.3. ORCA Computing PT-2 processor

The boson sampling system used in our experiments in section 4.2.3 is an ORCA Computing PT-2, which is a commercially available loop-based boson sampling system. It consists of a photon source based on parametric downconversion, two sequential optical delay lines of the same length, and photon-number resolving superconducting nanowire detectors. It supports up to 16 photons interfering in 32 time-bins. The coupling parameters between each optical delay line and the input/output channels are fully programmable.

Due to optical losses in the system, typically significantly fewer photons are detected than are sent in. As such, we also used post-selection where we populated all 32 input time bins and discarded all results in which fewer than 16 photons were measured. Since lost photons in a linear optical network act as though they had never existed in the first place, this regime effectively mimics roughly 16 photons in 32 channels with randomized input locations. Despite this input randomization, quantum effects such as photon bunching still occur and affect the output statistics.

To run our experiments, we collected a dataset of 500k samples with a fixed set of randomly initialized coupling parameters. Collecting this data took 40 minutes on this system. The system we used is hosted at the UK National Quantum Computing Centre.

## C. Experiments on 2D mixtures of Gaussians

These experiments used a WGAN-GP framework, where both the generator and discriminator are feedforward neural networks using two hidden layers with 256 neurons and LeakyReLU activations. All latent distributions were size-16. Both the distinguishable and indistinguishable samples were produced by randomly initialized simulated photonic systems with a 3-loop architecture in a 1-3-9 configuration. The models were trained for 5000 training steps, and all distributions were centered to have a mean value of 0 before being injected into the generator. Though figure 1 presents results for only a single seed, the experiments were repeated for 5 seeds with qualitatively repeatable behavior. When training for 10,000 iterations, we observed that the latent distribution with distinguishable photons was often able to close the gap with the indistinguishable photons, which implies that the generator is able to overcome the difference between those distributions on this dataset with further training. However, further training did not improve results with Gaussian and Bernoulli latent distributions. This is in line with the results from (Luise et al., 2020) on a similar toy dataset, where it was found that surprisingly deep neural networks were required to generate this dataset from an initial Gaussian distribution.

## D. Experiments on a synthetic quantum dataset

For our small-scale experiments, we use fully connected networks with 2 hidden layers of 512 neurons for both the generator and the critic. The generator output is of dimension 16. Both models use a ReLU activation function in all their hidden layers. All the latent distributions that we consider also have a dimension of 16. The training is done using batches of 500 samples over 40k iterations. As in (Gulrajani et al., 2017), we update the critic 5 times for each generator update. We use a RMSProp optimizer with a learning rate of $5 \times 10^{-4}$. The quantum latent distribution is obtained by sampling from a simulated boson sampler with 8 photons in an interference network with 16 channels. The interference network is mathematically described by a $16 \times 16$ unitary matrix, which we draw randomly from the Haar measure for each experiment. For each of the 12 experimental runs, we independently drew 3 random unitary matrices: one for the quantum latent distribution, one for the non-interfering photons, and one for the data the GAN is trained on.

# E. QM9 experiments

Our implementation started from the repository at https://github.com/kfzyqin/Implementation-MolGAN-PyTorch. We made a few changes to lead to improved performance. First, the MolGAN generator was modified by feeding affine transforms of the latent code **z** to all layers of the feedforward network, similar to the approach taken in (Karras et al., 2020). The activation functions used in the generator were also changed to LeakyReLU. Moreover, instead of using 3 layers, we used 5 layers with sizes 64, 176, 288, 400 and 512.

Data augmentation was used to extend the training dataset and make the model more amenable to translation and rotation invariance of molecules, which helped improve permutation invariance in the graph representation. A "permutation probability" parameter set to 0.3 was used to determine the strength of the data augmentation.

In all experiments, we used a batch size of 256 and the Adam (Kingma & Ba, 2015) optimizer with a learning rate of $10^{-4}$ and trained the models for 20000 iterations. Initially, the higher learning rate of $10^{-3}$ from the original repository was used, but we found that the lower learning rate of $10^{-4}$ led to significantly improved results.

The metrics were calculated with the code provided in https://github.com/MorganCThomas/MolScore. Example training curves for the size-16 latent distributions, with the FCD regularly evaluated during training, are shown in figure 4. Some examples of generated molecules can be found in figure 5

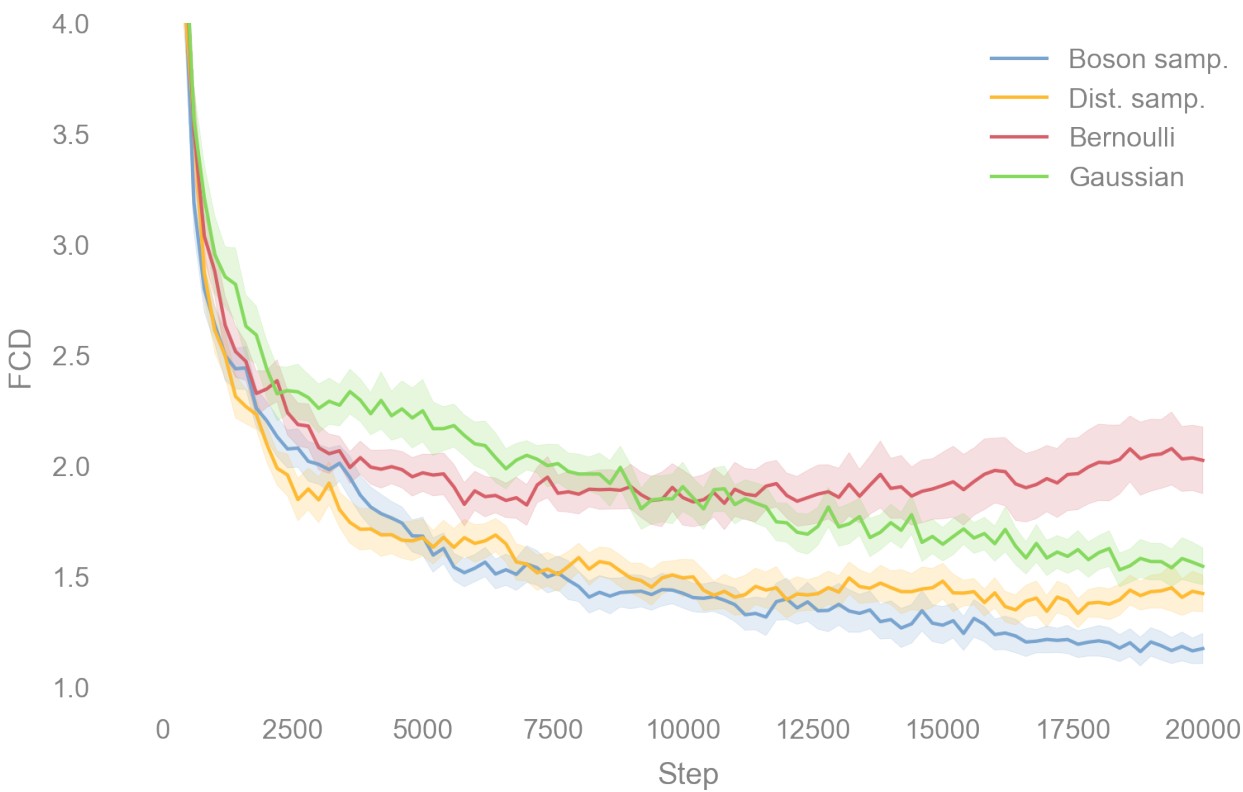

*Figure 4.* FCD scores during training for different latents of $d_z$=16. The mean and standard error of the mean are calculated from 20 random seeds.

## E.1. Note on circuit randomization

In general, to ensure that performance differences arise from general properties of the latent space distributions and not from specific circuit realizations, the optical circuits for both distinguishable and indistinguishable photons are re-sampled for every seed. This is the case for all experiments with the synthetic quantum dataset, and also for all size-16 and most size-32 latents on the QM9 dataset.

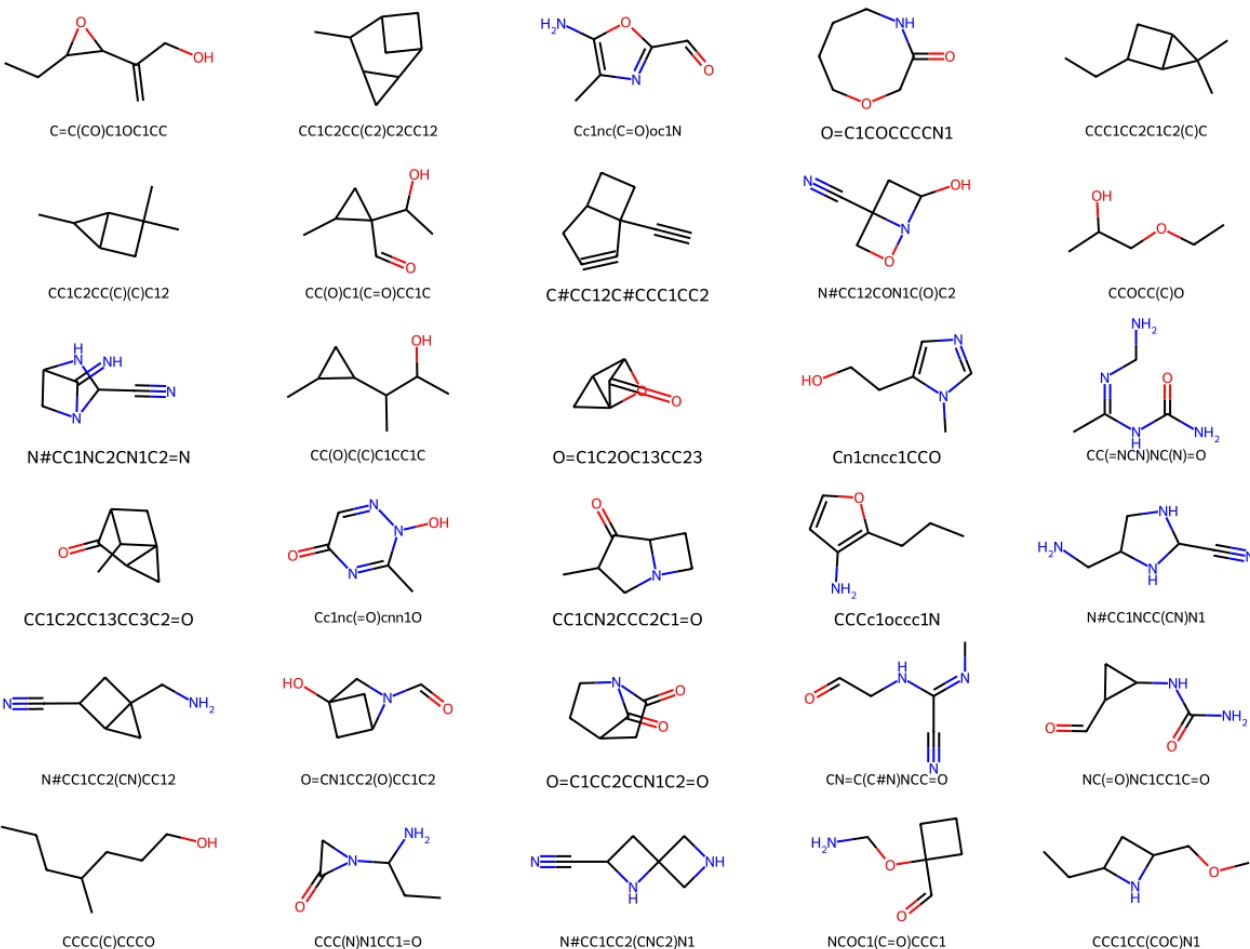

*Figure 5.* Examples of generated molecules using a trained model with a quantum latent distribution.

However, there are two instances in which the circuit was not re-randomized for every training seed, due to the time and computational requirements involved in collecting a sufficiently large number of samples to use as a latent distribution:

- Size-48 Haar-random circuits, for which collecting a single set of 1 million samples for indistinguishable photons took 3.75 hours on 100 CPUs. However, we observed from our experiments on size-32 Haar-random circuits that there is not much variability between the results obtained for different seeds for this type of circuit.

- Experiments using the ORCA PT-2 system, where collecting a single set of 500k samples took 40 minutes and experimental time was limited.

### E.2. Computational resources

The computational resources used for our QM9 experiments can be divided into the time taken to collect the latent space samples, and the training time. It took 40 minutes to collect 500k samples with an ORCA PT-2. For the simulated boson sampler size-32 models, 20 different sets of optical circuit parameters were used to precompute 1 million samples (a total of 20 million) and 1 million samples for the size-48 models using a single set of optical circuit parameters. The time taken to obtain these samples using a simulated boson sampler can be estimated using Table 5. The time taken to collect samples from simulated distinguishable photons was negligible.

Each training experiment ran on a single NVIDIA HGX™ A100 GPU 80GB. The size-16 models were trained using 20 different sets of optical circuit parameters where the samples were generated on the spot. For each of these types of models it took around 90 minutes to train when using the distinguishable sampler, 115 minutes for the boson sampler, and 48 minutes for both the Gaussian and Bernoulli distributions. Training the models for size-32 and size-48 and for both distinguishable and boson samplers (where we used a custom dataloader to load pre-computed samples) took around 55 minutes each. The time taken for other classical latents is the same as for size-16.

The tables below show the time and rates needed to generate simulated samples of a boson sampler with $n$ photons in $m$ optical channels.

*Table 5.* Sampling time for 500 samples on a single AMD EPYC™ 7003 processor

| n | m | Time |
|---|---|---|
| 8 | 16 | 19.6 ms $\pm$ 34.3 $\mu$s |
| 16 | 32 | 4.15 s $\pm$ 49 ms |
| 24 | 48 | 670 s $\pm$ 16.5 s |

## F. DDGANs

Diffusion models consider a process that gradually corrupts examples of a data distribution with noise. First, a forward diffusion process adds noise step by step, where this noise is sampled from a Gaussian distribution. The reverse process is where the generative process takes place. Given a fully corrupted image $\mathbf{x}_T$ and the current time step $t$, a neural network predicts the mean of a Gaussian distribution $\mu_\theta(x_t, t)$, such that sampling from that distribution removes the noise applied in one step.

Latent diffusion models generally make Gaussianity assumptions on the latent distribution (Rombach et al., 2022), such that the use of a quantum distribution is not straightforward. However, it has been observed that the reverse process, when considered over large time steps, involves modeling a complex non-Gaussian distribution, for which a GAN can be well suited (Xiao et al., 2021), therefore we focus on Denoising Diffusion GANs (DDGANs), which use the standard forward diffusion process but use a GAN to perform the reverse process. They achieve results comparable to standard diffusion models with faster inference speeds, with a model that is easier to train than other GANs. This work aligns with the hypothesis (Hu et al., 2023) that at least part of the observed improvement in diffusion models over GANs may stem from the additional computation at inference due to the recursive sampling process rather than a fundamental paradigm difference.

In DDGANs, a full multimodal conditional GAN is used at each step of the denoising diffusion process. The generative process is performed using 4 denoising steps ($T = 4$). For a given (fixed) Gaussian noise input $\mathbf{x}_T$, different images $\mathbf{x}_0$ are obtained using different samples from the latent distribution $z$. The combination of the initial Gaussian noise and samples from the latent determine the final result.

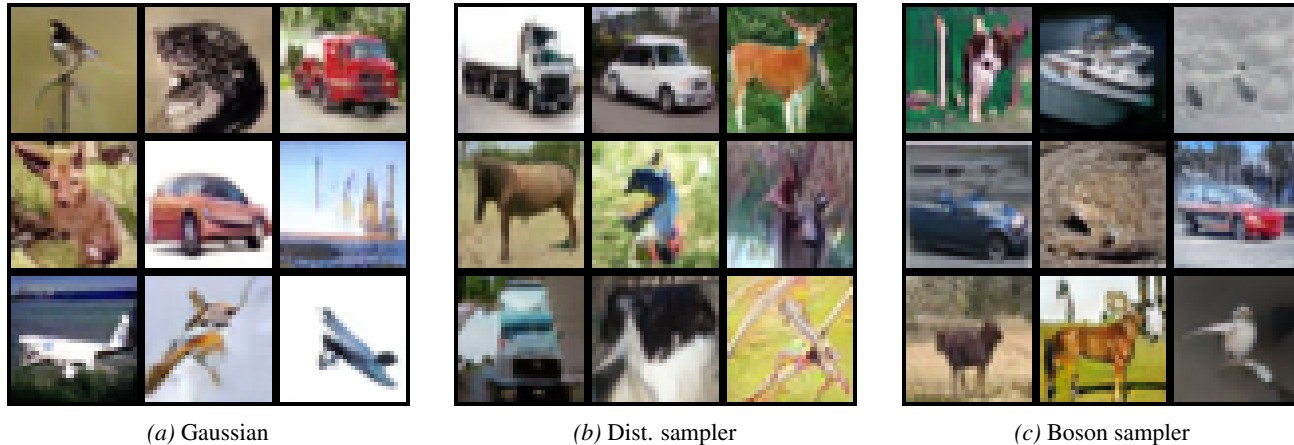

*(a)* Gaussian           *(b)* Dist. sampler           *(c)* Boson sampler

*Figure 6.* Some example images produced by our trained DDGAN models with three different latent distributions.

We investigated unconditional generation on the CIFAR-10 dataset (Krizhevsky et al., 2009) for different latent spaces, where the original model used a $d_z = 256$ latent space, our experiments used a $d_z = 48$ latent space due to the classical complexity of simulating larger quantum circuits. The quantum and distinguishable sampler latents for these experiments were produced by two unstructured optical circuits, with transfer matrices described by Haar-random unitary matrices, with 24 photons in 48 channels.

We used the code in `https://github.com/NVlabs/denoising-diffusion-gan` to run these experiments. The code was minimally modified to use the boson and distinguishable samplers as latents for the generator during training and inference. We used the commands in the repo to train and evaluate the models. Some example outputs are shown in figure 6. Though these experiments do not exhibit a performance advantage with a quantum latent distribution, they demonstrate the feasibility of this approach and motivate further exploration.

*Table 6.* Frechet Inception Distance (FID) for different latent distributions with a denoising diffusion GAN, with means and standard errors reported over 5 training seeds

| Latent Type | FID ↓ |
|---|---|
| Indistinguishable photons | $4.025 \pm 0.075$ |
| Distinguishable photons | $4.071 \pm 0.064$ |
| Gaussian | $4.040 \pm 0.040$ |

### F.1. Computational resources

Each training run took around 23.2 hours each, running on four NVIDIA HGX™ A100 GPU 80GB.

## G. Flow matching

To illustrate the feasibility of a quantum latent distribution as a source distribution, Figure 7 represents a small-scale model trained using the method from (Tong et al., 2023) to map 2D-projected quantum latent vectors to the "moons" dataset. We build on the Optimal-Transport variant of Conditional Flow Matching from (Tong et al., 2023), as implemented in `https://github.com/atong01/conditional-flow-matching`. To reconcile the dimensionality mismatch between the quantum latent distribution and our two-dimensional data, we introduce an untrained decoder that maps quantum latents to 2D source samples. Specifically, we first apply an affine transformation, scaling and shifting each latent vector by its global mean and variance, and then project the result through a Kaiming-normal–initialized kernel. This ensures that the resultant source distribution remains comparable in scale to the standard Gaussian source distribution. The decoder's parameters remain fixed throughout training, and all other training procedures follow the original codebase. During sampling, we draw from the quantum latent distribution, decode each sample, and then perform the usual flow-matching routine.

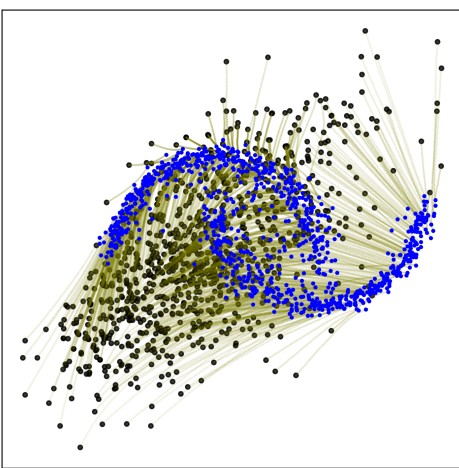

*Figure 7.* A flow matching model trained to map a set of quantum latent vectors projected onto a 2D plane (black) to the "moons" dataset (blue)

## H. Negative results

In this section, we present additional results for experiments in which we found that the performance of a GAN model was not improved by using a quantum latent distribution.

### H.1. StyleGAN with CIFAR-10

We investigated the StyleGAN model (Karras et al., 2019) with the CIFAR-10 dataset and different types of latent distributions. We used the inception score (Salimans et al., 2016) as the metric for model performance. Table 7 shows the inception scores (IS) (Salimans et al., 2016) achieved by the trained models. We find that the performance of all models is similar, suggesting that the choice of latent distribution matters less for this type of model and dataset.

*Table 7.* Inception scores (± standard error over 5 seeds) obtained on CIFAR 10 using different latent distributions.

|  | Gaussian | Bernoulli | Quantum |
|---|---|---|---|
| Inception score ↑ | $7.57 \pm 0.02$ | $\mathbf{7.66 \pm 0.04}$ | $7.54 \pm 0.01$ |

### H.2. QM9 experiments

We also found that small changes to the model and training regime used in our QM9 experiments could make a significant difference and erase a large part of the observed performance differences. Using the same model but with a higher learning rate of $10^{-3}$ (instead of $10^{-4}$), a permutation probability of 0.2, and 10,000 training steps (instead of 20,000) led to the results shown in table 8 for size-32 latents. There is no statistically significant difference between the latent distributions in this regime. The Gaussian distribution exhibits improved performance compared to the results from table 2, though still significantly worse than the best results using the size-16 latents. These results indicate that performance differences achieved using a different latent distribution are sensitive to the model and to the training parameters.

*Table 8.* Metrics (± standard error over 20 seeds) for different latents using training runs with a higher learning rate and a lower permutation probability

| Latent Type | $d_z$ | FCD ↓ | # Valid & unique ↑ | # Novel ↑ |
|---|---|---|---|---|
| Boson samp. 1-3-9 | 32 | $1.623 \pm 0.09$ | $1971 \pm 64$ | $1161 \pm 45$ |
| Dist. samp. 1-3-9 | 32 | $1.717 \pm 0.08$ | $1842 \pm 56$ | $1071 \pm 38$ |
| Gaussian | 32 | $1.554 \pm 0.06$ | $1756 \pm 58$ | $1048 \pm 42$ |

