# OpenReview forum: "Quantum latent distributions in deep generative models"
_ICML.cc/2026/Conference — ICML 2026 regular_

### Official Review · Reviewer_kJ4d · 2026-02-24

**Soundness:** 4
**Presentation:** 4
**Significance:** 3
**Originality:** 3
**Overall Recommendation:** 5
**Confidence:** 4

**Summary:**

The paper investigates the usefulness of using seeds from bosonic quantum computers for the task of generative modeling. For this purpose, the paper formalizes and proves intuitions regarding the classical post-processing of classically-hard probability distributions with artificial neural networks. Essentially, if the artificial neural network is invertible, and Lipschitz continuity bounds the changes between output and input distributions, then the output distribution is still classically hard to sample from. The paper provides benchmarks with and without multi-photon quantum interference, and compares them to Gaussian and Bernoulli seeds. While quantum improvements are shown also for some synthetic data sets, the most important quantum improvements are shown for a quantum chemistry problem. The generative models in the main text are all GANs, although the appendix also implements other variants of generative modeling methods with quantum seeds, as well as other datasets. However, the benchmarks in the appendix seem to be insensitive with regard to the choice of seed sampling process.

**Compliance With Llm Reviewing Policy:**

Affirmed.

**Final Justification:**

As commented in my rebuttal acknowledgement, my concerns have been addressed and I keep my positive score.

**Key Questions For Authors:**

1) Do the authors have estimates for the scaling of the quantum improvements? Table 2 seems to suggest that the performance gaps close with larger dimension. It could be that the quantum improvements are just an offset that becomes irrelevant for larger instances. However, it could also be that the gaps close because the task itself was not modified.

2) Why do the authors not seem to mention the benchmarks in the appendix that did not have quantum improvements? Again, I consider the research field to be in a very exploratory state, and quantum advantages are difficult to find. So, it is valuable information for other researchers to know on which tasks the method disappointed.

**Limitations:**

The discussion of the societal impact is short, but adequate. There are no particular impacts of this paper on society which go beyond the usual concerns of task-specific generative modeling with artificial neural networks or quantum computation.

As described above, I would appreciate if the authors could mention the appendix benchmarks more explicitly in the main text, because they inform other researchers about data sets which are less likely to profit from quantum seeds. That is valuable information.

**Strengths And Weaknesses:**

The formal part of the paper is sound, and results are robust against details of the formalization. The benchmarks are thorough, and meet expectations. I did not check the code itself, but the descriptions in the paper are reasonable.

The presentation is well-written and well-structured. The paper positions itself well within the literature, and thoroughly explains adaptations from other papers and code repositories. The paper presents its results in an appropriate, honest way. The authors might consider adding a sentence towards the end of the main text, mentioning that further benchmarks in the appendix show that some problems only show a negligible sensitivity to the choice of sampling process. This, however, was expected, and fits well together with the rest of the narrative. They might also point out that the so-called “photonic” seeds, while lacking quantum many-body interference, still have quantum interference in the form of single-photon self-interference.

The research field of generative artificial neural networks with quantum seeds itself is rather new, and therefore all formal results and benchmarks are helpful to guide the further development of the field. The main quantum competitors are fully-quantum machine learning, which requires large-scale error-corrected quantum computers, and variational quantum algorithms, which suffer from bad training landscapes. Therefore, finding other near-term quantum advantages in machine learning is crucial for the field, and the paper provides good arguments and evidence that combining quantum seeds with classical artificial neural networks is a promising direction to take.

In particular, the paper formalizes and proves that quantum hardness of sampling is robust to small modifications done by artificial neural networks, and this is reassuring to know. The numerics support the intuition that quantum seeds are most helpful on quantum problems (here, quantum chemistry), but find improvements for some synthetic data sets too. While the empirical improvements are not of a world-shaking nature, they are nonetheless helpful to further guide the field. In general, substantial quantum advantages are known to be very problem-specific. Therefore, also this research field profits from a thorough exploration of tasks and data sets, to which this paper contributes.

---

> ### Author Rebuttal · Authors · 2026-03-27
>
> We thank the reviewer for their time spent reviewing our paper.
>
> *Question 1 on scaling of the quantum improvements*: This is a valid point. The diminishing improvements could either reflect a diminishing advantage from using a quantum distribution at scale, or the task difficulty not scaling with latent dimension. Since there is no consistent improvement for the other latent distributions as the size increases, we feel larger latents of any type do not help for QM9.
>
> The best way to investigate scaling would be to extend this work to larger and more complex datasets (for example larger molecules or more complex chemical spaces) where larger latent spaces improve results. This is a natural next step, but significantly larger computational resources are needed to train more complex models with several different latents over multiple training seeds. We hope this paper can help motivate and encourage future work into this topic.
>
> *Question 2 on the benchmarks in the appendix*: We agree that sharing results on experiments where no quantum improvements were achieved is important to the field. This was not a deliberate omission from the main text, and we will make this more prominent by adding a couple of sentences in the Discussion section referring to the results in the appendix and contextualizing them.
>
> We will also amend the paper to point out that self-interference is still present in the “photonic” seeds, while highlighting the difference between self-interference and multi-photon interference.

---

> > ### Author Rebuttal · Reviewer_kJ4d · 2026-04-01
> >
> > I thank the authors for addressing my questions, which were suggestions rather than complaints. Since I already gave good scores before (in particular, a “5: Accept”), there is not really a point in raising the scores even higher.

---

### Official Review · Reviewer_G8QT · 2026-02-24

**Soundness:** 3
**Presentation:** 3
**Significance:** 3
**Originality:** 3
**Overall Recommendation:** 4
**Confidence:** 3

**Summary:**

The authors explore the performance of generative models making use of quantum latent distributions. The authors first provide theorems demonstrating that quantum latent variables can induce pushforward distributions that cannot be replicated by classical deep generative models. The authors then perform a number of benchmark tests using GANs based on quantum latent variables. The compare quantum GANs to GANs using Bernoulli, Gaussian and non-quantum "phontonic" distributions showing that for both synthetic datasets and the QM9 quantum chemistry dataset, quantum GANs outperform classical GANS in generations performance. The authors also compare a range of different quantum distributions as well as perform experiments using a real, rather than simulated quantum distribution.

**Compliance With Llm Reviewing Policy:**

Affirmed.

**Key Questions For Authors:**

- What is the support of the quantum distribution? Is it correct to understand it as a distribution over the counts of the, say 8, observed photons in each of 16 channels? What is the PMF of the "photonic" distribution, is it simply the sum of 8 i.i.d. random one-hot vectors?
- Are the L1 distances in table 1 comparable between the quantum and Bernoulli datasets? Again, based on my understanding these distributions would have different support.
- Why not try more complex classical distributions? E.g. for the example in figure 1, one might expect that a mixture of Gaussians latent should allow a model that closely matches the target distribution.
- GANs specified as the target model because they " provide a straightforward mapping from latent distribution to data distribution that facilitates apples-to-apples comparisons between latent distributions". It seems to me that the more important advantage of GANs is that they only require the ability to sample from the latent prior. Is it possible to train any DGM with quantum latents or is it restricted to models that only require sampling?
- Re: the beginning of section 4.2: "Generating chemical structures with specific properties is a challenging problem, partly because the underlying dynamics are described by quantum mechanical principles". Does the fact that chemical properties depend of quantum mechanical effects make the distributions of stable molecules quantum in the sense formalized in this paper? For your experiments on this dataset are you conditioning on properties?

**Limitations:**

As discussed above, I believe the authors could have better addressed the limitations of their work. Any work that makes progress toward generative modeling of chemicals with targeted properties should consider and address the risk of misuse.

**Strengths And Weaknesses:**

*Disclaimer: I had no knowledge several background topics prior to reading this paper particularly with regards to quantum distributions and quantum computing in general.*

**Strengths**
- The paper is clearly written and well-motivated, the methods introduced are theoretically sound. Finding applications for quantum computing in machine learning is a exciting line of work with potential for long-term impacts
- The paper provides simple, but important theorems demonstrating that "quantum" deep generative models can fit distributions that cannot be modeled by classical DGMs. The theorems appear to be sound as far as I can tell, though I have not thoroughly  examined the proofs given in the appendix.
- The authors provide both simulated and real testing of quantum deep generative models, demonstrating the effects of a quantum latent distribution in an apples-to-apples comparison with typical classical distributions commonly used in practice.
- The authors provide fairly extensive experiments on real-world datasets including the QM9 quantum chemistry dataset, which show the benefits of the quantum approach


**Weaknesses**
- The authors only compare to a very simple set of classical distributions making it unclear if the benefits truly come from the quantum nature of distributions considered. The authors simply argue that using quantum distributions is easier than testing a wider range of classical distributions.
- The authors don't address the cost and complexity of using quantum distributions, which I assume to be very high with current technology. In general the authors could better address the weaknesses of the proposed approach.
- While the writing is generally strong, the text is missing some important details about the distributions used (see questions below) leaving the reader to make inferences based on prior work.
- While some aspects of the evaluation are novel, the authors are not the first to introduce GANs based on quantum latent distributions, so the methodological novelty is limited.

---

> ### Author Rebuttal · Authors · 2026-03-27
>
> We thank the reviewer for their time spent reviewing our manuscript.
>
> In response to the reviewer’s questions:
>
> - The support for the quantum distribution is indeed the distribution over the counts of 8 photons in each of the 16 channels. As an example, with 3 photons in 6 channels possible samples could be [1,1,0,1,0,0] or [1,0,0,2,0,0]. In general, for N photons in M channels the number of possible samples is (N+M-1) choose N.
>   The photonic distribution has the same support, but its PMF is difference. Its PMF can indeed be described as a sum of independent one-hot vectors, where each one-hot vector indicates in which output channel a photon was detected. However, each input photon can have a different PMF of output one-hot vectors, since this depends on how each specific input channel is coupled to the output channels. We will add a clearer explanation of this in the paper.
> - This is a good point. The main focus of table 1 is on the comparison between columns. As the quantum and Bernoulli datasets do have different supports, the two rows are not exactly comparable. However, our metric compares the output of the GAN to the nearest integer on a per-channel basis, and on this basis 70% of the quantum dataset consists of 0s and 1s. The roughly 3x higher error on the quantum dataset is thus likely to reflect at least in part the higher complexity of this dataset, and not just the difference in support. We will add this qualification to the main text.
> - Our work aimed to provide a controlled experiment between different latent distributions. The comparison between quantum and photonic latents showed that quantum interference can be singled out as a single factor that improves performance. The comparison against Gaussian and Bernoulli distributions provides useful baselines to show that the quantum distribution can be competitive against standard choices for a latent distribution. Beyond those standard distributions, confounding factors can make an apples-to-apples comparison more difficult. For example, using a mixture of Gaussians would require specifying a number of modes in advance, which could make it a less general solution. Other work has also suggested trained classical distributions as latents, but we choose to focus on untrained distributions to ensure a fair comparison.
> - This is a valid point. The approach presented in our work is compatible with approaches that only require samples from the latent distribution. This includes the GANs that we focus on here, as well as some state-of-the-art flow matching models such as [Albergo 2025]. However, several DGM families, such as many current implementations of diffusion models or VAEs, make Gaussianity assumptions about the latent space that would make a quantum latent distribution a priori unsuitable. Though this point is mentioned in some parts of our paper (such as Appendix F on diffusion), we will further discuss this in the main text.
> - The fact that quantum chemical properties of molecules arise from quantum mechanics was a motivating heuristic for choosing to work with the QM9 dataset. It is not clear that the distribution of molecules in QM9 could be proven to be in any specific complexity class. Our experiments focus on distributional similarity between the generated molecules and the base molecules, and we do not condition on the properties. We will make these points clearer in the main text.
>
> The reviewer also raises some valid points about other weaknesses and limitations:
> - The cost and complexity of using quantum distributions is a fair point that is not sufficiently addressed in our paper. However, the practical barrier is lower than what many readers might assume. The quantum system we used is hosted at the UK National Quantum Computing Centre where it is available to researchers upon request. Other quantum systems are more directly available on the cloud via various providers. Though this is still not equivalent to widespread availability and costs can be a factor, as quantum computing matures we do expect that both cost and complexity for end user will decrease. This makes research into their potential ML applications timely. We will amend the paper to better discuss this.
> - We hope the additional information provided in our answers above, and which will be added to the paper, provide the missing details that were mentioned as a weakness.
> - Concerning the novelty of the methodology, we feel our theoretical contributions and our benchmarking of this approach are the primary novelty, rather than the use of quantum latents per se.
> - We will update the impact statement to better address the risk of misuse in generative modelling of molecules.

---

> > ### Author Rebuttal · Reviewer_G8QT · 2026-04-04
> >
> > Thank you to the authors for their response! The authors clarified most of my questions and still remain positive about this work as I think that it is interesting. However I still feel that there is a lack of a compelling argument that this approach would actually have practical benefits over alternative approaches, particularly given the complexity and cost of quantum sampling. Based on that my current score still reflects my opinion of the work.

---

### Official Review · Reviewer_PU5f · 2026-03-13

**Soundness:** 3
**Presentation:** 3
**Significance:** 3
**Originality:** 3
**Overall Recommendation:** 4
**Confidence:** 4

**Summary:**

The paper studies whether replacing the latent prior in deep generative models with a distribution produced by a quantum processor can yield output distributions that cannot be efficiently induced by classical latent distributions. The main theoretical claim is conditional: for generators with an efficiently computable inverse and Lipschitz regularity, a quantum latent distribution can induce output distributions that efficient classical latent samplers cannot reproduce. Empirically, the paper benchmarks GANs on a toy quantum dataset and on QM9 using Gaussian, Bernoulli, distinguishable-photon, and boson-sampling latents, including experiments on a real ORCA photonic processor.

**Compliance With Llm Reviewing Policy:**

Affirmed.

**Final Justification:**

This paper is more than incremental. It combines a theoretical argument, empirical benchmarking against multiple baselines, and real hardware results. However, the theoretical part is still not impacting me. Extensive benchmarking on both simulated and real photonic quantum processors is very helpful in the quantum machine learning community, but several baselines should be considered.

I highly evaluate the effort to discuss my questions in the rebuttal process. However,  understand that my questions (2-3) are not easy to address in a short rebuttal. The rebuttal does not change my evaluation. Therefore, I keep my current score as "Weak accept" as the final justification.

**Key Questions For Authors:**

Here are several minor comments:

1. The introduction currently does a good job motivating quantum latent distributions, but it could also include a short sentence clarifying that the paper does not claim a universal benefit over all classical latent modeling strategies. That would make the scope of the contribution more precise.

2. Why do the authors not consider a stronger classical baseline in which the latent prior is itself learned or transformed, rather than fixed to a simple parametric family? If not, please briefly justify in the experimental section why these particular classical latent families were chosen.

3. On QM9, what is the preferred interpretation of the gain: better inductive bias for chemistry-related data, greater latent expressivity, or specifically quantum-interference-induced structure?

4. The sections on diffusion and flow matching are interesting, but the paper should make it even more prominent that these are compatibility or feasibility demonstrations rather than strong performance validations.

**Limitations:**

Yes

**Strengths And Weaknesses:**

Overall, I believe this paper is more than incremental. It combines a theoretical argument, empirical benchmarking against multiple baselines, and real-hardware results. However, at first impression, I still do not see why quantum should help in general. The connection between theory and practice remains conditional and somewhat idealized. In my view, the paper supports only a weaker statement: quantum latents may provide more expressive and better-matched correlated priors than simple classical latents. Even so, it remains unclear why this mechanism should apply beyond datasets closely aligned with the statistics of the chosen quantum sampler.

For example, this argument is most compelling when the target data distribution is closely aligned with the structure of the quantum sampler, as in the synthetic quantum dataset. It is less convincing for QM9. In that case, the most plausible interpretation is that the quantum latent acts as a useful structured prior, rather than that quantum should help simply because molecules are quantum.


I would rate the significance as moderate within quantum machine learning and quantum generative modeling, but not field-changing for generative modeling as a whole. The paper makes a worthwhile conceptual contribution, but its practical and theoretical impact is still limited.

---

> ### Author Rebuttal · Authors · 2026-03-27
>
> We thank the reviewer for their time spent reviewing our paper.
>
> In response to the reviewer’s questions:
>
> 1. We agree it is important to make it clear that we are not claiming a universal benefit over all classical latent modeling strategies. The final sentence of the introduction is already intended to convey this idea. However, we take the reviewer's comment as a signal that this point could be rephrased or made more prominently, and we will revise accordingly.
> 2. While more complex classical distributions could also have been considered, our work aimed to provide a controlled experiment between different latent distributions. The comparison between quantum and photonic showed that quantum interference can be singled out as a single factor that improves performance. The comparison against Gaussian and Bernoulli distributions provide useful baselines to show that the quantum distribution can be competitive against standard choices for a latent distribution. Beyond those standard distributions, confounding factors can make an apples-to-apples comparison more difficult. For example, using a learned or transformed classical prior would introduce additional training complexity that would make it hard to distinguish between the contribution of the latent structure and that of the additional training, making a fair comparison difficult. We will add a brief justification of our choice of classical baselines to the experimental section.
> 3. This is an interesting question. We feel quantum-interference induced structure and inductive bias mechanisms are probably intertwined. In general, using a latent distribution that endows the model with the inductive bias that most closely aligns with the target distribution will lead to the best results. Quantifying this notion of inductive bias is however difficult, and the final model performance is likely to be the best proxy. The better results achieved by the quantum latent compared to the photonic latent thus implies that quantum interference creates structure that produces a more useful inductive bias for this dataset. Though the quantum latent is one way of producing this structure, it may not be the only way, but it is not clear which alternative approaches could be used in practice (without adding confounding factors such as training for example).
>   Given a target dataset, a successful approach could be to select a latent distribution that is a priori likely to yield a beneficial inductive bias. In our work, the underlying quantum mechanical physics of small molecules in chemistry acted as a motivating heuristic for us to select the QM9 dataset. \
>   We feel the notion of “latent expressivity” is a bit more ambiguous. Along the lines of “no free lunch” theorems, we feel it’s unlikely that one latent distribution will be the best or the most expressive in general. Selecting a latent distribution for its anticipated inductive bias for a model/dataset combination may be what works best in practice, though it is not always clear how such a choice should be made.
> 4. We agree, and will make this more prominent
>
> In connection to question 3 above, in the “Strengths and Weaknesses” section the reviewer raises a general point about the gap between the theoretical conditions and the practical results. We agree that the conditions of Theorem 1 are somewhat idealized, and that practical benefit of quantum latents will often reflect their role as structured priors rather than a computational separation from classical distributions. However, Theorem 1 does provides a principled theoretical basis for why the quantum nature of the distribution is not simply destroyed by the neural network transformation, which is a result that was previously lacking in the literature and which we feel is a useful contribution even though the theoretical conditions can not always be met. Moreover, even when the practical mechanism is best understood as 'quantum latents provide a useful structured prior,' we feel demonstrating this through controlled experiments — in particular the apples-to-apples comparison between distinguishable and indistinguishable photons — is itself a meaningful contribution. We will revise section IIID of the paper and the Discussion to make this interpretation more explicit.

---

> > ### Author Rebuttal · Reviewer_PU5f · 2026-04-03
> >
> > I thank the authors for the rebuttal effort. I understand that my questions (2-3) are not easy to address in a short rebuttal. The theoretical part is still not impacting me. However, we highly evaluate the effort to discuss our question. Furthermore, extensive benchmarking on both simulated and real photonic quantum processors is very helpful in the quantum machine learning community.

---

### Official Review · Reviewer_jKyt · 2026-03-14

**Soundness:** 2
**Presentation:** 3
**Significance:** 2
**Originality:** 2
**Overall Recommendation:** 3
**Confidence:** 1

**Summary:**

This paper asks whether distributions from quantum processors can serve as better latent distributions in generative models like GANs. The idea is that the generator has finite capacity, so a more structured latent distribution that's closer to the data can ease the generator's job. Quantum distributions from photonic boson sampling have rich correlations and non-factorizable structure that simple latents like Gaussians lack.
The paper contributes a theoretical result showing that quantum latents can produce output distributions that classical latents cannot even approximate under certain generator conditions, intuition for when this should help in practice, and benchmarks on a synthetic dataset and the QM9 molecular dataset.

**Compliance With Llm Reviewing Policy:**

Affirmed.

**Key Questions For Authors:**

- How would you expect quantum latent distributions to perform against alternatives like copulas or normalizing flows that can also capture rich correlations and multimodal structure?

- Beyond datasets arising from quantum-mechanical processes, can you characterize more precisely which types of problems or datasets would benefit from quantum latent distributions? Is there a practical way to predict in advance whether a quantum latent would help for a given task?

**Limitations:**

yes

**Strengths And Weaknesses:**

I have limited background in this area, but my impression is:

Strengths:
The main strength is that the core idea seems natural: explore the potential of quantum latent distributions.

Weaknesses:
- The classical baselines seem potentially limited? The paper compares against Gaussian, Bernoulli, and distinguishable photons, but these are all relatively simple distributions. Could an expressive classical distribution like a copula or a normalizing flow achieve similar correlations and multimodal structure? The paper acknowledges this gap but dismisses it by saying testing many alternatives is impractical. So I'm not sure whether they have adequately motivated the need for quantum latent distributions.

- The approach is only demonstrated on GANs, which are no longer state of the art for most generative tasks. The appendix shows feasibility for diffusion and flow matching but without performance comparisons, so it's unclear whether the findings transfer to modern architectures.

---

> ### Author Rebuttal · Authors · 2026-03-27
>
> We thank the reviewer for their time spent reviewing our paper.
>
> In response to the reviewer’s questions:
> - On the theoretical side, copulas and normalizing flows belong to complexity class C, so Theorem 1 still provides a theoretical basis for why a quantum latent can induce output distributions that these approaches cannot efficiently reproduce.
>
>   On the empirical side, while more expressive classical distributions such as copulas or normalizing flows could be considered, our work aimed to provide a controlled experiment between different latent distributions. The comparison between quantum and photonic showed that quantum interference can be singled out as a single factor that improves performance. The comparison against Gaussian and Bernoulli distributions provide useful baselines to show that the quantum distribution can be competitive against standard choices for a latent distribution. Beyond those standard distributions, confounding factors can make an apples-to-apples comparison more difficult. Using copulas or normalizing flows would introduce additional training complexity that could make it hard to separate the contribution of the latent structure with that of the additional training.
>
>   We note that quantum distributions can also in principle be included into more complex strategies, where for example the parameters of the quantum processor are also trained jointly with the neural network. Expanding our work to incorporate these strategies would then justify a comparison against more complex classical strategies. Though this is beyond the scope of our current work, we hope this paper can help motivate and encourage future work into this topic.
>
> - This is a good question. When not using a default distribution that provides decent results (often Gaussian), the choice of latent distribution is often guided by a mix of heuristic arguments and past empirical performance. To help with this decision, our work provides heuristic arguments in section IIID, suggesting that quantum latent distributions can help provide an inductive bias for datasets arising from quantum processes or for highly non-factorizable data. Beyond these heuristics, the comparison between photonic and quantum performance on a given dataset could itself serve as a diagnostic: if quantum interference helps over distinguishable photons, the dataset likely has structure that benefits from quantum statistics. However, we acknowledge that predicting in advance whether a quantum latent will help for a given task remains an open question, and we flag this as an important direction for future work.
>
> Concerning the limitation pointed out by the reviewer that our work focuses on GANs, these were chosen deliberately for benchmarking because they provide a direct mapping from latent distribution to data distribution, making comparisons between latent distributions cleaner. The theoretical result in Theorem 1 applies to any generator architecture satisfying the stated conditions, not just GANs. Further extending this approach to these models is a promising direction for future work that we hope to motivate using the results in this paper, including the feasibility demonstrations in the appendix.
>
> We hope these answers help address the reviewer's questions and concerns.

---

> > ### Author Rebuttal · Reviewer_jKyt · 2026-04-05
> >
> > I thank the authors for their response; I think I'm just not fully convinced about the practical motivations or applications of the proposed methods and so I maintain my score.

---

### Decision · Program_Chairs · 2026-04-30

**Decision:**

Accept (regular)

**Comment:**

The submission addresses an interesting question in quantum machine learning: whether quantum-generated latent distributions can enhance deep generative models, and how such gains might relate to genuinely quantum properties of the latent distribution. As noted by Reviewer PU5f, the work is more than incremental. In particular, the combination of theoretical motivation, benchmarking across multiple settings, and experiments on both simulated and real photonic quantum processors makes the paper a useful contribution to the area.

The main strengths of the paper lie in its breadth and overall execution. The empirical study is reasonably extensive, includes real-hardware experiments, and goes beyond a narrowly scoped proof of concept. That said, the paper is not without weaknesses. The theoretical component is not yet fully convincing or especially impactful, and the practical case could be articulated more sharply, especially given the cost and complexity of quantum sampling. Additional baselines would also have further strengthened the empirical evidence.

On balance, I believe the paper makes a sufficiently interesting and well-supported contribution to warrant acceptance.